# Proton gradients and pH oscillations emerge from heat flow at the microscale

Lorenz M. R. Keil[1], Friederike M. Möller[1], Michael Kieß[1], Patrick W. Kudella[1] & Christof B. Mast[1]

Proton gradients are essential for biological systems. They not only drive the synthesis of ATP, but initiate molecule degradation and recycling inside lysosomes. However, the high mobility and permeability of protons through membranes make pH gradients very hard to sustain in vitro. Here we report that heat flow across a water-filled chamber forms and sustains stable pH gradients. Charged molecules accumulate by convection and thermo-phoresis better than uncharged species. In a dissociation reaction, this imbalances the reaction equilibrium and creates a difference in pH. In solutions of amino acids, phosphate, or nucleotides, we achieve pH differences of up to 2 pH units. The same mechanism cycles biomolecules by convection in the created proton gradient. This implements a feedback between biomolecules and a cyclic variation of the pH. The finding provides a mechanism to create a self-sustained proton gradient to drive biochemical reactions.

---

[1] Systems Biophysics, Physics Department, Center for Nanoscience, Ludwig-Maximilians-Universität München, 80799 Munich, Germany. Correspondence and requests for materials should be addressed to C.B.M. (email: christof.mast@physik.uni-muenchen.de)

The establishment of ion gradients holds an important place for the sustainment of all current life forms. Regions of different pH are essential for various cellular processes such as the synthesis of DNA, RNA, and proteins as well as their degradation and recycling[1–4]. These processes are accommodated in specialized compartments separated by biological membranes and important in lysosomes, mitochondria, and cell vacuoles. To form complex reaction networks, the variation of pH is essential to create different reaction conditions[5, 6]. In the process of chemiosmosis, for example, the channeled movement of ions across a membrane drives the phosphorylation of ADP to ATP, life's most commonly used energy currency[7]. Since ancient microbes, such as the last universal common ancestor (LUCA), are assumed to be chemiosmotic[8], it has been argued toward the significance of proton gradients at the origin of life[9–11]. In addition, the synthesis of precursor molecules for early molecular evolution requires differing pH values for the synthesis of purine and pyrimidine ribonucleotides[12], aminonitriles[13], amino acids[14], and phosphoenol pyruvate[15].

For living systems, the creation of pH differences is tedious and requires complex protein-mediated transport mechanisms. On prebiotic earth, natural pH gradients would form by laminar mixing of fluids with different pH values, e.g., at the interface between disequilibria at submarine hydrothermal vents and the Hadean ocean[16–18]. However, these systems rely on a continuous influx of mass and energy. Molecules would experience this gradient only for a short time and are washed out into a pH-equilibrated reservoir, keeping the mechanism transient. A repetitive pH oscillation inside a closed system would be desirable and keep molecules at one location without being lost by a flow or by diffusion.

Here we found a mechanism to form stable pH gradients and continuous cycling of pH in a closed system. Thermal energy, the form of energy necessary for the mechanism, triggers an accumulation mechanism for individual ionic species. The heat flow forms a temperature gradient, spanning across sub-millimeter-sized water-filled compartments. This is likely a common setting on early earth, found for example in geothermally heated porous rocks such as hydrothermal vents or cooling volcanic sites[19, 20]. This elementary setting has previously been shown to concentrate dilute nucleotides[21], accumulate lipids to facilitate the formation of vesicles[22], shift polymerization reactions toward longer DNA/RNA strands[23], and selectively replicate longer polymers[24]. This study now extends these characteristics to include the formation of a stable pH gradient by thermally separating dissolved buffer molecules of different charge states. This locally shifts the buffer equilibrium, yielding pH differences of up to two units and provides another aspect for a promising long-term microhabitat for the onset of molecular evolution.

In the following experiments, we demonstrate the formation of proton gradients in various buffer solutions. These include prebiotically plausible solutions of phosphate buffers, amino acids, and nucleotides[25–29]. In a temperature gradient, the thermophoretic movement of each ionic species is different. As a result, the concentration of ionic species is locally shifted, prompting the formation of proton gradients (Fig. 1). As indicated by the modeling, the formation of proton gradients is not affected by smaller concentrations of large biomolecules or vesicles. Such entities are shuttled inside the already established pH gradient by laminar convection, prompting them to move regularly between regions of varying pH. Thereby, each molecule undergoes individual pH oscillations with respect to their molecular properties, mostly their diffusion coefficient and Soret coefficient. The pH oscillations enable chemical reaction pathways that rely on pH fluctuations. The comparably fast shuttling of vesicles could trigger proton gradients across protocellular membranes without the need for active proton pumps.

## Results

**Thermally driven accumulation.** The formation of pH gradients is facilitated by a thermally driven accumulation mechanism inside a water-filled compartment. These flow chambers are subjected to a spatially confined horizontal temperature gradient. Based on the second law of thermodynamics, energy fluxes are necessary to maintain states of low entropies, in this case accumulated regions of molecules. The accumulation mechanism relies on the superposition of two effects: (i) thermal convection drives molecules up- and downwards in the flow chamber and (ii) thermophoresis—the movement of molecules in a temperature gradient (Fig. 2a). In combination, the two effects result in an efficient net transport of molecules, as small as single ions, to the bottom of the flow chamber and a simultaneous depletion at the top. The flux $j_i$ for an ionic species $i$ inside the flow chamber is composed of diffusion, thermophoresis, and convection. It can be described by

$$\mathbf{j}_i = -D_i \nabla c_i - S_{Ti} D_i \nabla T c_i + \mathbf{v} c_i \qquad (1)$$

Where $\mathbf{v}$ denotes the convection flow, $c_i$, $D_i$, and $S_{Ti}$ the concentration of the ionic species $i$, diffusion coefficient, and Soret coefficient, respectively[23]. The Soret coefficient $S_{Ti} = D_{Ti} D_i^{-1}$ is defined as the quotient of the thermal and collective diffusion coefficient. Using Debye's approach, an analytical solution for the accumulation mechanism in a flow chamber at equilibrium is given by[30, 31]

$$\frac{c}{c_0} = \exp(\kappa_i S_{Ti} \Delta T r) \qquad (2)$$

where $c/c_0$ denotes the relative concentration of a species at the bottom of the chamber, $c_0$ the initial concentration, the temperature difference $\Delta T$, the aspect ratio of the chamber $r = h\, w^{-1}$ with the height $h$ and width $w$, and the experimental prefactor $\kappa_i$ ranging between 0 and 0.42 (see Eqs. 10 and 11). The experimental prefactor $\kappa_i$ is calculated for each ionic species and depends on experimental parameters such as the temperature difference $\Delta T$, the chamber width $w$, and the diffusion coefficient of the species $D_i$.

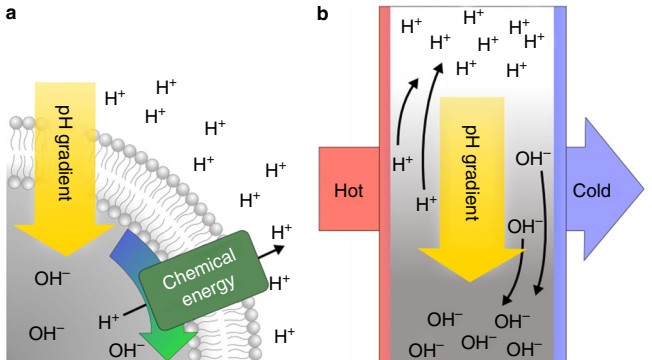

**Fig. 1** Formation of pH gradients by chemical or thermal energy. **a** Modern cells run on an elaborate protein system to maintain proton gradients across a membrane. Chemical energy is used to pump protons against their concentration gradient which is then harnessed by chemiosmosis. **b** Heat fluxes across confined solutions, a common geological setting, induce a movement of ions that results in a stable pH gradient of up to 2 pH units. The charge selective thermophoretic accumulation of buffer molecules such as amino acids, phosphates, or nucleotides keeps the water self-ionization persistently out of equilibrium. The formation of a pH gradient is achieved in a closed system and biomolecules are repetitively exposed to differences in pH

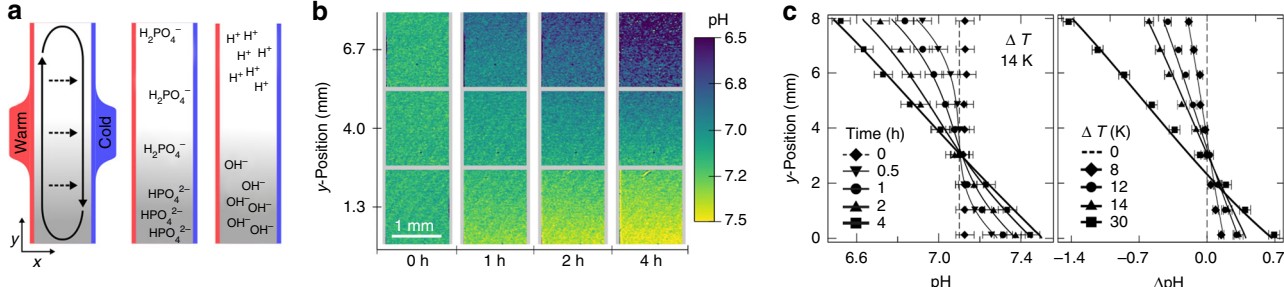

**Fig. 2** Formation of pH gradients in phosphate solutions. **a** The temperature gradient creates a convective flow in the solution while simultaneously inducing a movement of molecules along the temperature gradient based on thermophoresis. The combination of both effects results in an accumulation of molecules at the bottom of the chamber. For the case of the phosphate buffer, $HPO_4^{2-}$ accumulates much stronger compared to $H_2PO_4^-$. Based on the concentration of both buffer species, the pH is locally shifted. As a result, the pH increases at the lower regions and decreases at the upper regions of the chamber. **b** The formation of a pH gradient is monitored in a 8 mm high chamber via the ratiometric dye SNARF-1. **c** For a given temperature difference of $\Delta T = 14$ K, a stable proton gradient of $\Delta pH = 0.93$ develops within 4 h (see Supplementary Movie 1). The steepness of the proton gradient changes with respect to $\Delta T$ and is in good agreement with our theoretical model. Differences in pH are achieved after 4 h for a $\Delta T = 8$, 12, 14 K and after 1 h for a $\Delta T = 30$ K. The error bars give the standard deviation of the SNARF-1 detection for duplicate measurements

For a given buffer reaction in solution, each buffer subspecies accumulates differently. Its respective accumulation efficiency, as shown in Eq. (2), strongly depends on molecular properties which define its diffusion and Soret coefficients. As a result, different ratios in concentrations of proton acceptor and donor species are present along the height of the flow chamber, which then form a pH gradient (Fig. 2b, c). Chamber widths above 150 μm for Earth's gravity field are optimal for the accumulation of fast diffusing, small molecules to achieve a maximum pH gradient. These pores exhibit high convection velocities, which are necessary to counterbalance the fast diffusion of the molecules. As a result, the accumulation mechanism concentrates small molecules at the bottom of the pore quickly. Longer cannot compensate for the convection on the considered experimental and simulation timescales. Until they reach their steady state of thermophoretic accumulation after days or even weeks, they shuttle through the thermal habitat and the zones of different pH. The Soret coefficient of the phosphate buffer species are estimated with respect to their net charge (see Supplementary Table 2)[32].

**Phosphate-mediated formation of stable pH gradients**. At a pH of 7, phosphate buffer consists mostly of two ionic species, hydrogen phosphate ($HPO_4^{2-}$) and dihydrogen phosphate ($H_2PO_4^-$). Both ionic species obtain different vertical accumulation profiles along the flow chamber, as the dominant contribution to the thermophoretic force approximately scales with the charge of the molecules squared (Fig. 3a). Based on their net charge, $HPO_4^{2-}$ accumulates tenfold stronger compared to $H_2PO_4^-$, locally shifting the acid–base equilibrium in the buffer solution. The acid–base reaction in general and for the phosphate buffer solution in particular is defined by

$$HX + H_2O \rightleftharpoons X^- + H_3O^+ \quad (3)$$

$$H_2PO_4^- + H_2O \rightleftharpoons HPO_4^{2-} + H_3O^+ \quad (4)$$

where HX and $H_2PO_4^-$ denote the proton donors and $X^-$ and $HPO_4^{2-}$ the proton acceptors. High concentrations of $HPO_4^{2-}$ at the bottom of the chamber drive a protonation reaction that locally reduces the hydronium ion concentration (Fig. 3b). Simultaneously, a deprotonation reaction of highly concentrated $H_2PO_4^-$ releases hydronium ions at the top of the chamber. The acid–base reaction locally shifts the pH along the chamber. The pH of a solution is derived by the Henderson–Hasselbalch equation

$$pH = pK_a + \log\left(\frac{c_{acceptor}}{c_{donor}}\right) \quad (5)$$

where $c_{acceptor}$ and $c_{donor}$ denote local proton acceptor and proton donor concentrations, respectively. Given sufficiently high buffer concentrations, the pH is only affected by the ratio of proton acceptors to donors, and not by their total concentrations. The pH at the bottom of the chamber $pH_{bottom}$ is therefore derived by calculating the corresponding proton acceptor/donor concentration at the bottom, given by Eq. (2) at equilibrium.

$$pH_{bottom} = pK_a + \log\left(\frac{c_{0\,acceptor}}{c_{0\,donor}} \exp(\Delta(\kappa_i S_{Ti})\Delta Tr)\right)$$
$$= pK_a + \log\left(\frac{c_{0\,acceptor}}{c_{0\,donor}}\right) + \frac{\Delta(\kappa_i S_{Ti})\Delta Tr}{\ln(10)} \quad (6)$$

with $\Delta(\kappa_i S_{Ti}) = \kappa_{acceptor} S_{T\,acceptor} - \kappa_{donor} S_{T\,donor}$ and $r = hw^{-1}$ the aspect ratio of the chamber. By assuming $\kappa_{acceptor} \approx \kappa_{donor}$ and $1/\ln(10) \approx 0.43$, $pH_{bottom}$ is given by

$$pH_{bottom} \approx pH_{init} + 0.43\,\kappa\left(S_{T\,acceptor} - S_{T\,donor}\right)\Delta Tr \quad (7)$$

The maximum pH difference $\Delta pH$ inside the flow chamber, arising between the lowest and highest regions of the chamber, is given by

$$\Delta pH \approx 0.86\,\kappa(\Delta S_T)\Delta Tr \quad (8)$$

with $\Delta S_T = S_{T\,acceptor} - S_{T\,donor}$. The steepness of the pH gradient is affected by the temperature difference $\Delta T$, Soret coefficient $S_T$, and height $h$ and width $w$ of the chamber (Fig. 3c and Supplementary Fig. 4). An elongated flow chamber can compensate for lower temperature differences, achieving similar or higher pH gradients (Fig. 3d). The theoretical model is applicable for shallow proton gradients and is in good agreement with experimental data such as the formation of pH gradients in solutions of phosphate. Here the model predicts a proton gradient of $\Delta pH = 0.40$, 0.90, and 1.2, which describes comparably well the experimental values of $\Delta pH = 0.34$, 0.69, and 0.93 for a given temperature difference of $\Delta T = 8$, 12, and 14 K, respectively (Fig. 3c). The pH model (Eq. 8), however, neglects the build-up of an electrical field that leads to an electrophoretic movement of ions, inhibiting the formation of a pH gradient. As a result, the

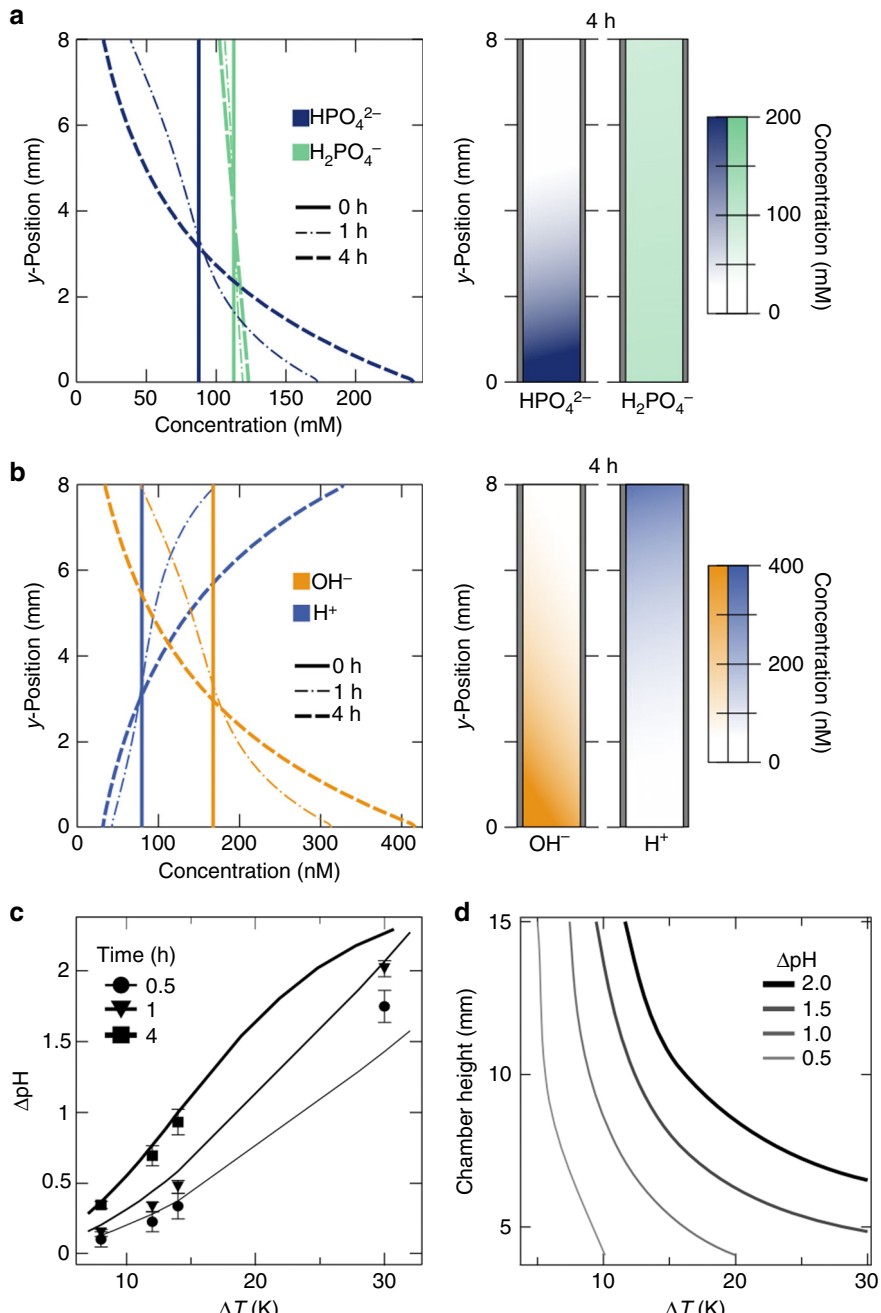

**Fig. 3** Accumulation of phosphate buffer species imbalances the reaction equilibrium. **a** Each phosphate species accumulates at the bottom of the chamber, depending on its thermophoretic force. At near neutral pH, $HPO_4^{2-}$ and $H_2PO_4^-$ are predominately in the solution. As the thermophoretic force scales with the charge of the molecules squared, $HPO_4^{2-}$ achieves a tenfold higher accumulation compared to $H_2PO_4^-$. **b** The protonation reaction of $HPO_4^{2-}$ reduces the hydronium ion concentration at the bottom of the chamber while the deprotonation reaction of $H_2PO_4^-$ simultaneously releases hydronium ions at the top. The system reaches its steady state on the order of hours. **c** The pH gradient scales approximately linearly with the applied temperature difference. A maximum proton gradient of $\Delta pH = 2.1$ is formed for $\Delta T = 30$ K. **d** An elongated flow chamber compensates for lower temperature differences, achieving similar or higher pH gradients. The error bars give the standard deviation of the SNARF-1 detection

model is not applicable for steep proton gradients, e.g., formed by large differences in temperature or by large differences in Soret coefficients of proton donors/acceptors. In addition, above formula is subjected to the limitations of the Henderson–Hasselbalch equation, i.e., limited to acid/base concentration in the millimolar regime and pH values of about 7. Better descriptions of the experimental results are obtained using time-dependent finite-element simulations (Fig. 3c), including explicit reaction kinetics for the self-

ionization of water, acid–base reaction kinetics, and electrostatics (see Supplementary Tables 1–9). Simulation results are derived using the time-dependent finite-element model with a commercial solver.

**Ionic solutions mediate formation of pH gradients**. The formation of a proton gradient was not limited to phosphate buffer solutions but resulted also from various ionic and amino acid

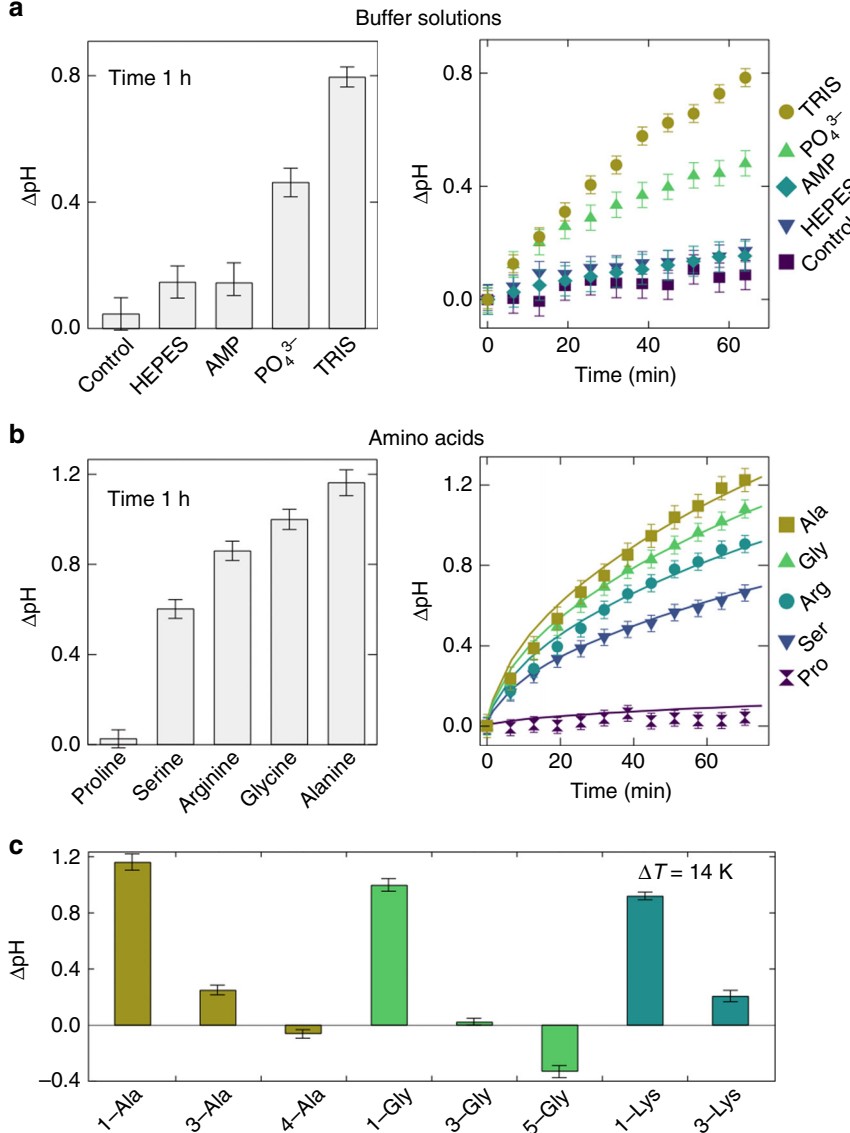

**Fig. 4** Formation of proton gradients in ionic and amino acid solutions. **a** Various ionic solutions form steep proton gradients within 60 min, achieving $\Delta pH$ = 0.14, 0.14, 0.47, and 0.79 for HEPES, AMP, phosphate, and TRIS, respectively. As a control, the fluorescent pH indicator SNARF-1 is dissolved in pure water, forming a pH gradient of $\Delta pH$ = 0.09. **b** Prebiotically plausible amino acids form proton gradients of up to $\Delta pH$ = 1.1 within 60 min. Finite-element simulations are in good agreement using differences in Soret coefficients $\Delta S_T$ of 0.022, 0.019, 0.016, 0.012, and 0.003 $K^{-1}$ as fitting parameters for the acceptor and donor species of alanine, glycine, arginine, serine, and proline, respectively. **c** Measured peptides have the tendency to form an inversed proton gradient, showing a decrease of pH at the bottom. An inversion of pH gradients is attributed to a negative difference in Soret coefficient $\Delta S_T$ from non-ionic thermophoresis. Error bars give the standard deviation of the SNARF-1 detection

solutions (see Supplementary Figs. 5–12). The primary requirement is an acid–base reaction and therefore buffer solutions of HEPES, TRIS, single nucleotides formed stable pH gradients (Fig. 4a). The fluorescent pH indicator SNARF-1 could also act as a buffer, however, due to the low concentrations of 50 µM, only shallow pH gradients emerged as shown in the control measurement. Solutions of TRIS and phosphate achieve the highest pH gradients of $\Delta pH$ = 0.47 and 0.79 at concentrations of 200 mM after 60 min. Amino acids have the capability to react with both bases and acids. Prebiotically plausible amino acids, such as alanine, arginine, glycine, proline, and serine, at concentrations of 200 mM formed proton gradients ranging between $\Delta pH$ = 0.1 and 1.1 within 60 min (Fig. 4b). Finite-element simulations describe the formation of the pH gradient in detail by assuming a difference in Soret coefficient $\Delta S_T$ of 0.022, 0.019, 0.016, 0.012, and

0.003 $K^{-1}$ for the acceptor and donor species of alanine, glycine, arginine, serine, and proline, respectively. These Soret coefficients are in the range for unevenly charged ionic species[32]. Our models indicate that the thermophoretic properties of each buffer species are the major source for differences in the proton gradients. However, the thermophoretic properties of small molecules are rather tedious to measure and not all available, making it difficult to give a detailed prediction for individual buffers[33]. Mixtures of various buffer or amino acids are expected to form an averaged pH gradient, derived from Eq. 2 and seen on the example for phosphate/arginine mixtures (see Supplementary Fig. 8). This pH gradient, however, is shifted by the exact ionic composition of the solution (see Eqs. 12–14). Interestingly, short chains of amino acid monomers, shown for tetra-glycine and penta-alanine, form an inverted proton gradient of $\Delta pH$ = −0.33 and −0.06,

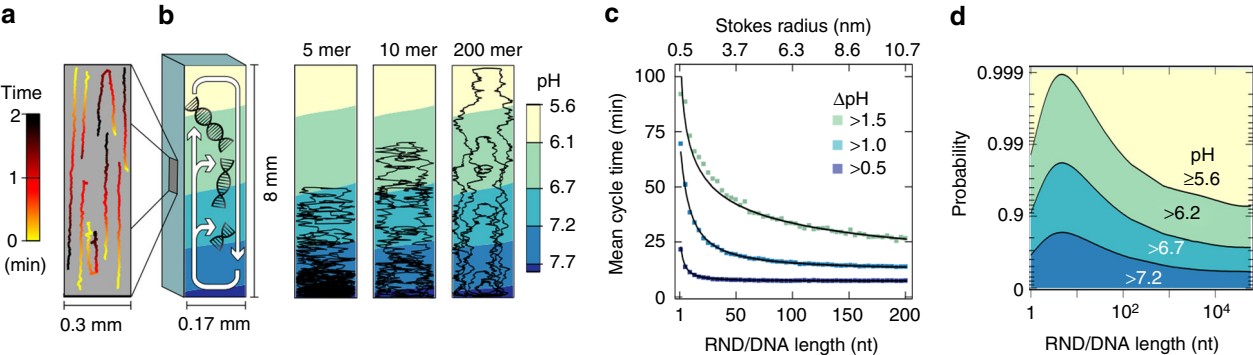

**Fig. 5** pH oscillations of individual particles derived by numerical simulations. **a** Laminar convection induces an upward and downward movement of molecules inside the chamber. The convection flow is visualized with 1 μm polystyrene beads over the time of 2 min (see Supplementary Movie 2). **b** Single molecules are shuttled in the existing pH gradient, undergoing subsequent pH oscillations. The more stochastic trajectories of a 5, 10, and 200-mer RNA or DNA strand are depicted in black. The probability for a molecule to remain in a specific pH region strongly varies for each molecule. **c** Stochastic simulations of 1–200 mer oligomers of RNA/DNA are used to derive mean cycle times for completing pH oscillation of ΔpH > 0.5, 1.0, and 1.5. Longer strands of RNA/DNA are more prone to complete a full pH cycle and therefore exhibit faster cycle times compared to shorter strands. **d** Short DNA/RNA strands accumulate quickly at the bottom, remaining mostly in a distinct pH region. Longer strands, however, are shuttled between varying pH regions for days until reaching the steady state of accumulation. The thickness of the flow chamber thereby determines which molecules remain at defined pH regions and which undergo frequent pH oscillations (see Supplementary Fig. 13 for details)

respectively (Fig. 4c). Here the accumulation of proton donors is much stronger compared to proton acceptors, resulting in a deprotonation reaction of proton donors that releases hydronium ions at the bottom of the chamber. The inversion of the pH gradient is explained by a negative difference in the Soret coefficient $\Delta S_T = S_{T\ acceptor} - S_{T\ donor}$, which is the result of additional non-ionic contributions to thermophoresis. The non-ionic contribution $S_T^{NI}$ of the Soret coefficient $S_T$ (see Eq. 12) scales linearly with the polymer length, as a result of local, molecule-solvent interaction around the tube-like polymer[32, 34, 35]. In the case of longer peptides, the contribution of the Soret coefficient which depends on the protonated state $S_T^{CM}$ becomes comparatively smaller. As a result, the difference in Soret coefficient $\Delta S_T$ between the proton acceptor and donor decreases or is even inverted for longer peptides.

**Convectively driven pH oscillations.** Molecules and particles accumulate and undergo laminar convection. The molecular motion inside the chamber is described by convection, Brownian motion, and thermophoresis (Fig. 5a, see Supplementary Movie 2). As a result, molecules shuttle in the arising pH gradient and are subjected to subsequent pH oscillations. These pH oscillations strongly depend on the molecular properties. Larger macromolecules are more prone to shuttle in the convective flow and exhibit faster pH cycle times compared to smaller molecules, which tend to solely accumulate at the bottom of the chamber (Fig. 5b). For example, 10-mer strands of RNA complete pH oscillations of at least ΔpH = 0.5, 1.0, and 1.5 in 11, 38, and 73 min, respectively (Fig. 5c). Longer DNA/RNA strands such as 200 mers undergo faster pH oscillations of at least ΔpH = 0.5, 1.0, and 1.5 in 7, 14, and 26 min, respectively.

The accumulation mechanism not only defines the magnitude of pH oscillations for a species, but also affects the probability for a species to remain in a specific pH region (Fig. 5d). The flow chamber efficiently accumulates DNA/RNA strands in the region of 3–10 nt, while smaller and larger molecules accumulate significantly weaker within the considered timescales. A 10-mer oligonucleotide inside an already formed pH gradient ranging from 7.7 to 5.6 has probabilities to stay in pH regions pH > 7.2 and pH > 6.7 of 0.81 and 0.98, respectively. This probability

decreases for shorter and longer oligonucleotides. The accumulation kinetics hereby strongly depends on the flow chamber's width; smaller chambers achieve larger accumulation for longer molecules (see Supplementary Fig. 13). The inversion of the pH gradients, facilitated by longer peptides, allows for a microscopic chemical feedback from polymerization back to the pH gradient in these chambers. Now, long molecules that are concentrated at the bottom of the chamber remain at low pH values for prolonged periods if the pH gradient is inverted.

## Discussion
Our experimental and theoretical findings show that heat fluxes across confined solutions form and sustain stable pH gradients. The mechanism solely requires a dissolved acid–base reaction buffer that is given by prebiotically plausible biomolecules such as phosphate, amino acids, or RNA. Phosphate for example is assumed to play an important role in prebiotic chemistry because it functions as a nucleophilic catalyst, pH buffer, and chemical buffer[29]. The formation of pH gradients is hereby achieved in a closed system that allows molecules to undergo repetitive pH oscillations without being washed away into an equilibrated reservoir.

The dissipation of free thermal energy, the sole form of energy necessary, is a geologically abundant scenario on early Earth. Although the formation of pH gradients depends on the thermal gradient that drives the accumulation mechanism, shallow thermal gradients can be compensated by longer flow chambers[36]. The geometry of the flow chamber is realized to mimic the simplest geometrical setting with dimensions of 8.0 × 0.17 mm. However, the shape of the flow chamber is found to have a minor impact on the accumulation mechanism[21].

The thermally driven formation of pH gradients can be incorporated into a microfluidic device that forms tunable pH gradients and subjects molecules to pH oscillations defined in range and frequency. This device potentially allows for size selection and certain replication mechanisms to be functional[24].

Interestingly, the existing pH gradient can simultaneously be utilized by actively driven biomolecules, not only based on Brownian motion. The accumulation mechanism thereby subjects molecules to convectively driven pH cycles, resulting in

permanent pH oscillation. Importantly, the oscillatory behavior applies for each individual species of the solution and strongly depends on its molecular properties. Hereby, specific molecular species harness the pH gradient by undergoing extensive pH oscillations, while others strongly accumulate by remaining in a limited pH region. This enables the formation of biomolecules that rely on fluctuations in pH during their assembly. The yield of pH-dependent multicomponent reaction systems, such as a three-component purine/pyrimidine reaction system, could be shifted along the height of the chamber. Here, purine ribonucleotides precursors would be synthesized at the top of the chamber at pH 5 while pyrimidine precursors would be synthesized at the bottom at pH 7[12].

An exciting prospect of this work is the periodical formation of proton gradients across vesicles. The leakiness of the first membranes[37], made of fatty acids or phosphorylated isoprenoids[38, 39], ensures a short-term pH equilibration with its surrounding. Based on the convective flow, vesicles are transported into different pH regions while preserving the equilibrated pH value at the inside, forming pH gradients across the membrane. These pH gradients could be harnessed by primordial forms of the ATP synthase to drive an energy storage system[11, 40].

Our results conclusively show in both theory and experiment how heat flows form stable pH gradients. Proton gradients hold an important place for the sustainment of all current life forms and have also been argued to play a central role in powering the origin of life[9, 10]. The simplicity of this mechanism opens various possibilities for rebuilding biological systems bottom up—to either understand them from a mechanistic perspective or to elucidate how cellular life might have originated.

The mechanism that forms pH differences here also drives the accumulation of dilute biomolecules[21, 24], overcoming the concentration problem of the origin of life. The combination of such an accumulation mechanism that simultaneously subjects biomolecules to pH oscillations prompts interesting future experiments. In the origin of life, the abundance of geological environments that can form stable pH gradients in a closed system could have opened vast chemical possibilities for the formation and sustainment of the first replication cycles of prebiotic biomolecules.

## Methods

**Microfluidic flow chamber**. The temperature gradient is applied across a 3D-printed flow chamber, in between a silica wafer and a transparent sapphire window (Sappro, Germany, see Supplementary Fig. 1). Hereby, a water bath (CF41 Kryo-Kompakt-Thermostat, Julabo, Germany) cools down the silica wafer while simultaneously heating up the sapphire using temperature PID-controlled heater cartridges (12 V, 6/20 mm, RepRap, France, see Supplementary Fig. 2). These heater cartridges are inserted into copper blocks that are thermally connected to the sapphire window. The silica wafer (Si-Mat-Silicon Materials e.K., Germany) has a diameter of 100 mm, a thickness of $525 \pm 25$ μm, a 100 nm $SiO_2$ coating, and a < 100 > orientation. The boundaries of the flow chamber are fabricated by a 3D printer using polylactic acid (PLA) material (Filamentworld, Germany). The flow chamber is positioned using a translation stage (Thorlabs, Germany). For in- and outflow of the chamber, capillaries with a height and width of $50 \times 500$ μm (vitrotubes 5003, CMScientific, UK) connected to a BTP E-60 tubing ($0.76 \times 1.22$ mm, Instech Laboratories, Inc., USA) are used. A high precision syringe pump (neMESYS, Cetoni, Germany) and 250 μL microsyringes (Hamilton, USA) ensure an accurate loading of the flow chamber at flow rates ranging from 1000 to 1 nL s$^{-1}$.

**Ratiometric pH detection**. The pH in the solution is derived using the ratiometric dye SNARF-1 (5-[and-6]-Carboxy SNARF-1, Invitrogen AG, CA) at concentrations of 50 μM (see Supplementary Fig. 3 and Supplementary Table 10). The starting pH of each solution was adjusted close to the pK of the SNARF-1 dye to ensure maximum accuracy of the pH measurement. For recording the fluorescence images at $\lambda_1 = 580$ nm and at $\lambda_2 = 640$ nm simultaneously, a self-built fluorescence microscope was used with a CMOS Stingray 145-b camera (AVT, Germany), a x2 objective (LD, NA 0.055, Mitutoyo, Japan) and an Optosplit 2 (Cairn Research, UK). The ratiometric dye is excited using a 470 nm LED (M470L2-C4, Thorlabs, Germany) in combination with a ratiometric filterset (BrighLine HC 482/35,

HCBS506, Brightline HC 580/23, H606LP, BrightLineXF 643/20, AHF Analysentechnik AG, Germany). Depicted error bars are given from duplicate measurements ($n = 2$), and in particular cases from triplicates.

**Finite-element simulations**. The finite-element simulations consist of three consecutive calculations of a two-dimensional flow chamber. The flow chamber is mimicked by using a rectangularly shaped compartment with a horizontal aligned temperature gradient.

The temperature profile of the flow chamber is calculated using partial differential equations for transient heat transfer. Thereby, a temperature of $T_{max}$ and $T_{min}$ is applied to the left and right vertical boundaries, respectively.

The convective flow profile inside the flow chamber is derived by numerically solving the incompressible Navier–Stokes equation. The micrometer-sized width and a moderate temperature gradient lead to a Rayleigh number of $Ra_L = 2$. As a result, the heat flow as well as the temperature profile almost entirely corresponds to heat transfer by conduction. The heat transfer caused by convection are negligible. Finite-element simulation of the heat transfer and fluid flow were preformed consecutively to decrease the computational time, however, yielded the same results compared to a coupled simulations of heat transfer/fluid.

Molecular concentrations for each buffer species are derived by simultaneously solving acid–base reaction kinetics, electrostatics, and molecular flux equations. Following Debye's approach, the simulation neglects perturbations at the upper and lower reversal points by assuming, where $v$ denotes the flow velocity. Explicit reaction rates and kinetics are listed below for both cases, solutions of phosphate buffer and amino acids (see Supplementary Tables 1–9).

**pH detection/calibration method**. A spatially resolved pH is derived using the ratiometric dye SNARF-1. Following a modified Henderson–Hasselbalch equation (see Eq. 9)[41], the fluorescence intensity ratio $R = F^{\lambda 1}/F^{\lambda 2}$ provides a measure for the pH. The wavelengths $\lambda_1$ and $\lambda_2$ denote the emission wavelength of SNARF-1 at $\lambda_1 = 580$ nm and $\lambda_2 = 640$ nm. The fluorescent images are recorded simultaneously using a beam-splitting device and background correction.

$$pH = a + b \log\left(\frac{R - Ra}{Rb - R}\right) \quad (9)$$

The modified Henderson–Hasselbalch equation comprises of four parameters ($Ra$, $Rb$, $a$, and $b$), derived from the ratio-to-pH calibration curve (see Supplementary Fig. 3). The parameters $a$ and $b$ denote for the $pK_a$ of SNARF-1 and the weighting of the spectra parameters while $Ra$ and $Rb$ account for the maximum and minimum fluorescence rations. Temperature-dependent shifts of the pH calibration curve are corrected by linearly fitted parameters $a$, $b$, $Ra$, and $Rb$ (see Supplementary Fig. 3 and Supplementary Table 10). If not mentioned separately, shown measurements were conducted in 50 μM SNARF-1, a buffer/amino acid concentration of 200 mM and a temperature gradient of $\Delta T = 14$ K.

**Experimental prefactor κ**. The experimental prefactor for the accumulation ratio is defined by[31]

$$\kappa = \frac{q/120}{1 + q^2/10080} \quad (10)$$

with $q$ defined as

$$q = \frac{\Delta T \alpha g \rho_0 w^3}{6 \eta D} \quad (11)$$

where $\Delta T$ denotes the temperature gradient, $\alpha$ the volume expansion of the solvent, $g$ standard gravity, $w$ the width of the chamber, $\rho_0$ the density of the solvent, $\eta$ the viscosity of the solvent, and $D$ the diffusion coefficient of the species. The accumulation prefactor strongly affects the accumulation efficiency. Therefore, the width of the flow chamber is specifically designed for certain molecules.

**Thermophoresis**. The thermophoretic effect moves molecules along a temperature gradient, mostly from the warm to the cold side. However, thermophoresis is still subject to active research. The Soret coefficient gives a measure for a molecules thermophoretic movement. It can thereby be described by four contributions, the capacitor $S_T^{CM}$, Seebeck effect $S_T^{EL}$ [42, 43], non-ionic $S_T^{NI}$, and the temperature contribution[32].

$$S_T = S_T^{CM} + S_T^{EL} + S_T^{NI} + 1/T \quad (12)$$

Recently, it has been suggested that thermophoresis on charged molecules is dominated by Seebeck effects and ion shielding[32]. The Seebeck effect solely depends on the ionic composition of the solution. Here each ion in solution is subjected to thermophoresis and generates a global electric field. Within the electric

field, molecules are subjected to electrophoresis. The electrical field is given by

$$E = \frac{k_B T \nabla T}{e} \frac{\sum_i Z_i c_i S_{Ti}}{\sum_j Z_j^2 c_j} \quad (13)$$

where $E$ denotes the electric field, $Z_i$, $c_i$, and $S_{Ti}$ denotes the charge number, concentration, and Soret coefficient of an ionic species. The Seebeck contribution can be approximated by

$$S_T^{EL} = -\frac{k_B T \mu_{DNA}}{e D_{DNA}} \frac{\sum_i Z_i c_i S_{Ti}}{\sum_j Z_j^2 c_j} \quad (14)$$

where $\mu_{DNA}$ denotes the electric mobility. As shown in Supplementary Fig. 7, the Seebeck contribution strongly affects the formation of pH gradients.

**Charge of amino acids and peptides**. The net charge of a peptide sequence or single amino acid depends on the pH and is given by[44]

$$Z = \sum_i N_i \frac{10^{pK_{ai}}}{10^{pH} + 10^{pK_{ai}}} - \sum_j N_j \frac{10^{pH}}{10^{pH} + 10^{pK_{aj}}} \quad (15)$$

where $Z$ denotes the net charge of the peptide sequence, $N_i$ the number of histidine, lysine, and arginine residues, and the N terminus with their respective $pK_{ai}$ values, and $N_j$ the number of cysteine and the C terminus with their respective $pK_{aj}$ values.

The net charge of an amino acid changes with respect to the alpha-amino group ($pK \approx 9$), the carboxylic acid group ($pK \approx 2$) and in some cases the side chain. The initial pH is adjusted to be situated in the detection range of SNARF-1 (pH 5–9), therefore solely the dissociation of the alpha-amino group and side chain are detectable (see Supplementary Fig. 8).

**Random walk model**. Random walk simulations investigate the pH cycling statistics of RNA/DNA strands. These molecules are placed inside a $0.17 \times 8.0$ mm flow chamber filled with 200 mM of phosphate solution. In theory and experiment, a temperature difference of $\Delta T = 30$ K forms a pH gradient of up to $\Delta pH = 2.1$ (see Figs. 2c and 5a). The particle movement ($N = 1000$) inside the already established pH gradient is simulated for a given time-step of $\Delta t = 1$ ms for a total simulation time of 3 h. The displacement $\Delta s(x, y)$ of these particles is given by

$$\Delta s(x, y) = \sqrt{4 \cdot D \cdot \Delta t} \cdot \eta(t) + \Delta t \cdot (\mathbf{v}(x, y) + D \cdot S_T \cdot \nabla \mathbf{T}(\mathbf{x}, \mathbf{y})) \quad (16)$$

Where $\Delta t$ denotes the time-step, $\mathbf{v}(x, y)$ the convective flow profile, and $T$ the local temperatures. Finite element simulations calculate the convective flow profile (see chapter finite-element simulations) while the Brownian motion is implemented by a randomly directed movement $\beta(t)$. Diffusion and Soret coefficients for RNA/DNA strands in the range of 1–200 nt are given by $D = 470 \times n^{-0.53} \mu m^2 s^{-1}$ and $S_T = 0.1 \times n^{0.5} K^{-1}$, where $n$ denotes the length of the strand. The experimental scaling law of the Soret coefficient $S_T$ is adapted from Mast et al.[23].

**Data availability**. The data that support the findings of this study are available from the corresponding author on reasonable request.

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

## Acknowledgements

We thank Matthias Morasch, Victor Sojo, and Dieter Braun for discussions and valuable comments on the manuscript. Financial support from the NanoSystems Initiative Munich, the Simons Collaboration on the Origin of Life, the Ludwig-Maximilians-Universität Munich Initiative Functional Nanosystems, and the SFB 1032 Project A4 is acknowledged.

## Author contributions

L.K., M.K., P.K. and C.B.M. performed the experiments. L.K. and F.M. performed the simulations. L.K. and C.B.M. conceived and designed the experiments, analyzed the data, and wrote the paper.

## Additional information

**Competing interests:** The authors declare no competing financial interests.

