## [Peer Review File · Nature Communications]

Reviewers' comments:

Reviewer #1 (Remarks to the Author):

This article by Keil et al demonstrates how pH gradients can form within small, millimetre sized chambers containing small molecules with pH titratable groups through convection and thermophoresis. The Braun lab has pioneered the construction and exploitation of thermophoretic chambers for a variety of applications, including the investigation of the origin of life. The pH was monitored by fluorescence with the ratiometric pH sensor fluorophore SNARF-1. pH gradients that spanned up to 2 units were measured. The authors then go on to model how such conditions could lead to nucleic acids experiencing oscillations of pH.

The work makes the important point that under the right conditions, pH gradients can spontaneously emerge, which may have been important for the origin of life. However, in my opinion, the work is a bit preliminary. A few more experiments would significantly improve the manuscript.

Comments:

"Also, regions of different pH are essential for various cellular processes such as the synthesis of DNA, RNA, and proteins as well as their degradation and recycling.²⁻⁵"

This statement and the references seem to be referring to eukaryotic cells. If an origin of life argument is being made, comparisons with eukaryotic cells does not seem to fit.

"The creation of pH gradients in a closed system however is difficult, especially without the help of complex protein-mediated transport mechanisms."

I don't follow the point. Cells are not closed systems.

"In the following experiments, we demonstrate the formation of proton gradients in various buffer solutions. These include prebiotically plausible solutions of phosphate buffers, amino acids, and RNA.²¹⁻²"

Experiments with RNA were only simulated.

"Solutions of TRIS and phosphate achieve the highest pH gradients"

An explanation why would be helpful to the reader.

Why does Ala give the largest difference in pH? It's intuitive that Pro would be the worst, but not that Ala would be the best. The Soret coefficients reflect the chemical properties of the molecules, right? So it would be helpful to the reader to explain things a bit more chemically rather than just stating Soret coefficients.

"The inversion of the pH gradient is explained by a negative difference in the Soret coefficient $\Delta S_T = S_T \text{ acceptor} - S_T \text{ donor}$, which is the result of additional non-ionic contributions to thermophoresis." What are these "additional non-ionic contributions?" What are the implications of longer peptides inverting their distribution?

Why do larger molecules cycle throughout the chamber and smaller molecules gets stuck at the bottom?

There is no data to confirm the simulations of oscillatory, convectively driven RNA cycling between different pH regions of the chamber. Demonstrating that the described system actually works in the lab would greatly improve the impact of the paper. Presumably, it shouldn't be so difficult to monitor this with fluorescently labelled RNA molecules.

What happens if mixtures of amino acids are present in the chamber? Would the pH gradient be lost? Since mixtures of molecules were likely present on prebiotic Earth, it is important to understand how robust the system is to mixtures.

Although error bars are reported (which is good), there is no indication of the number of replicates.

An argument is made that the pH gradient could be exploited by a protocell, but no clear explanation of how is made.

Reviewer #2 (Remarks to the Author):

PROTON GRADIENTS EMERGE FROM HEAT FLOW AT THE MICROSCALE

Keil et al.

This is a rigorous and convincing demonstration that temperature gradients concentrate solutes differentially and produce pH gradients in closed systems. The results are clear and the analysis sound to the best of my understanding.

Given the importance of this process for the origin of metabolism the paper is of wide importance.

I do not require it, but given the broad range of phenomena mentioned in the introduction, the authors might wish to mention plant cell vacuoles as well as lysosomes. The authors might also form a view on the possible relevance of the exergonic reactions of mitochondrial respiration forming temperature gradients within cells:-

Chrétien et al. 2017 Mitochondria are 1 physiologically maintained at close to 50 °C. bioRxiv doi: <http://dx.doi.org/10.1101/133223>

PDF attached

Small points.

I would somehow specify "across biological membranes" in the first sentence (line 14). pH gradients occur on a macro scale, too, in animals and plants, and it might help the general reader to focus from the beginning on cells, membranes and membrane compartments.

Line 26 "hold" should be "holds".

Reviewer #3 (Remarks to the Author):

The paper builds on the authors' prior work describing thermophoresis effects on molecules in confined geometries ("pores"), which is a body of work in the greater goal to understand conditions that allowed biochemical reactions in the early formation of life. The authors are leading this area and have firmly established the effect for concentration of biomolecules and particles of widely varying size and properties, and they have shown how these conditions affect some biochemical reactions including synthesis of long nucleic acids. The current manuscript makes a significant advance beyond prior work by demonstrating formation of stable pH gradients – alongside biomolecule transport/concentration, pH gradients could provide pH cycling conditions that enabled synthesis reactions before the origin of cells with active transport mechanisms. While in some ways this work is a natural evolution in the research of the authors, elucidation of pH gradients in a plausible natural system (heated pore in a rock) is a fundamentally important advance in the grand scheme. In addition, the work is high of quality and complete (with a few small suggestions below).

Main questions:

1. Throughout this paper, long DNA is less confined/trapped than short oligonucleotides (3-10 nt). This seems opposite of findings in the authors' previous work (e.g., Figure 2 of "Extreme accumulation of nucleotides in simulated hydrothermal pore systems") where it appeared that longer DNA was confined/trapped more effectively or as effectively. The authors should comment on this apparent difference in findings.
2. Flow conditions and modeling. The Supplementary Material shows that the model was done as three consecutive simulations, but it is not clear how the first two simulations (thermal and fluid flow) can be de-coupled. Specifically, the thermal distribution should be affected by convective heat transport (flow), and the temperature-induced density differences (thermal) should be required to model the resulting fluid flow. In addition, it would be helpful to have brief comment on the scaling of buoyance-driven flow compared to the dimensions/properties used here (e.g., via Rayleigh number).
3. The paper presents a simplified model and notes that a more complete model was developed – for all simulated results, it should be clarified which model was used. Also, the simplified model can be useful for understanding the key parameters in a simpler context – if the simulation results presented are for the full model, it would be useful to comment on how the simplified model relates (Does it give similar predictions for points that can be related to the simulations? Is the simplified model valid across the range of conditions reported? What are the limitations in range of applicability of the simple model?)

Minor questions/clarifications:

4. Line 47: "thin sheet of water". This phrase might imply a film with an open face, rather than an enclosed volume – it would be better to reword this for accuracy and also include the essential elements of a temperature gradient.
5. Line 47: "this elementary and abundant setting". It seems that the authors intended this to refer specifically to a pore with a thermal gradient, but it is not clear. It may be helpful to describe why this condition is elementary and abundant – it is explained further below, but it is not explained here.
6. Line 50: "these characteristics now include...". It was not clear to me (until looking at the references) if this was referring to work that had already shown pH gradients created by thermal gradients in pores (like the work here) or some other work. It looks like these articles describe the role of pH gradients in early biochemical reactions, but they are created by an "un-natural" laboratory method, so the meaning of this sentence should be clarified.
7. Line 58: "larger or less concentrated molecules or vesicles". Meaning and origin of the statement is not clear – "larger" I can understand (from the results here), but I don't understand "less concentrated" in this context (less concentrated small molecules?).
8. Line 87: "diffusion" to "diffusion coefficient"
9. Line 93: it may be helpful to briefly note the most relevant parameters included in the experimental prefactor.
10. Figure 2. Scale bar should be included, and it is not clear what region of the chamber are shown in each colored panel. Missing temperature units for 30 near end of caption.
11. Line 180: "4c). Here, hydronium ions are released at the bottom of the flow chamber while simultaneously reducing them at the top."
12. Line 182: "which is the result of additional non-ionic contributions to thermophoresis." It is not clear what this means.
13. Supplementary Information: the models results are included in some figures but not in all – there is plenty of verification of results here so it is not a critical point, but if there is a reason models were not included in some cases, it would be good to comment. If not, it would further strengthen the work to include them.
14. Figure caption 4c: "c, Peptides have the tendency to form an inversed proton gradient, showing a decrease of pH at the bottom." Is this generally true or only for the specific peptides observed?
15. Results section for Figure 5 (around lines 198-220): It seems that the text is out of line with figure

order, and some references to figures seem incorrect.

- “Stochastic simulations show that 10-mer strands of RNA complete pH oscillations of at least $\Delta\text{pH} = 0.5$, 1.0, and 1.5 in 11, 38 and 73 min, respectively (Fig. 5b).” – intended to refer to Figure 5d or 5c?
- “As a result, longer DNA/RNA strands such as 200mers undergo faster pH oscillations of at least $\Delta\text{pH} = 0.5$, 1.0, and 1.5 in 7, 14, and 26 min, respectively.” – intended to refer to Figure 5c?
- “The accumulation mechanism not only defines the magnitude of pH oscillations for a species but also affects the probability for a species to remain in a specific pH region (Fig. 5a).” – Figure 5b??
- “The flow chamber efficiently accumulates DNA/RNA strands in the region of 3-10 nt, while smaller and larger molecules accumulate significantly weaker (Fig. 5c).” – Figure 5b??
- End of paragraph, and Figure 5 caption – intended to refer to Fig S13, not S12?

Barry Lutz
Associate Professor
Department of Bioengineering
University of Washington

Reviewers' comments:

Reviewer #1 (Remarks to the Author):

This article by Keil et al demonstrates how pH gradients can form within small, millimetre sized chambers containing small molecules with pH titratable groups through convection and thermophoresis. The Braun lab has pioneered the construction and exploitation of thermophoretic chambers for a variety of applications, including the investigation of the origin of life. The pH was monitored by fluorescence with the ratiometric pH sensor fluorophore SNARF-1. pH gradients that spanned up to 2 units were measured. The authors then go on to model how such conditions could lead to nucleic acids experiencing oscillations of pH.

The work makes the important point that under the right conditions, pH gradients can spontaneously emerge, which may have been important for the origin of life. However, in my opinion, the work is a bit preliminary. A few more experiments would significantly improve the manuscript.

Comments:

“Also, regions of different pH are essential for various cellular processes such as the synthesis of DNA, RNA, and proteins as well as their degradation and recycling.^{2–5}”

This statement and the references seem to be referring to eukaryotic cells. If an origin of life argument is being made, comparisons with eukaryotic cells does not seem to fit.

We rewrote the starting paragraph to differentiate more clearly between pH gradients in today's life and potential implication on the origin of life. We now write in the resubmitted manuscript: “The establishment of ion gradients holds an important place for the sustainment of all current life forms. Regions of different pH are essential for various cellular processes such as the synthesis of DNA, RNA, and proteins as well as their degradation and recycling. These processes are accommodated in specialized compartments separated by biological membranes and important in lysosomes, mitochondria, and cell vacuoles. To form complex reaction networks, the variation of pH is essential to create different reaction conditions. In the process of chemiosmosis, for example, the channelled movement of ions across a membrane drives the phosphorylation of ADP to ATP, life's most commonly used energy currency. Since ancient microbes, such as the last universal common ancestor (LUCA), are assumed to be chemiosmotic, it has been argued towards the significance of proton gradients at the origin of life. In addition, the synthesis of precursor molecules for early molecular evolution requires differing pH values for the synthesis of purine and pyrimidine ribonucleotides, aminonitriles, amino acids, and phosphoenol pyruvate.”

“The creation of pH gradients in a closed system however is difficult, especially without the help of complex protein-mediated transport mechanisms.”

I don't follow the point. Cells are not closed systems.

We apologize that this point was not clear enough. We now write: “For living systems, the creation of pH differences is tedious and requires complex protein-mediated transport mechanisms.”

"In the following experiments, we demonstrate the formation of proton gradients in various buffer solutions. These include prebiotically plausible solutions of phosphate buffers, amino acids, and RNA.^{21–2}"

Experiments with RNA were only simulated.

More precisely, we also meant parts of RNA, i.e. nucleotides. We demonstrate the formation of proton gradients in solutions of AMP (see Figure 4a) as well as dAMP, ATP and dCMP (see Supplementary Fig. S12). We are now more explicit on this point in the revised manuscript by writing: “These include prebiotically plausible solutions of phosphate buffers, amino acids, and nucleotides”.

"Solutions of TRIS and phosphate achieve the highest pH gradients"

An explanation why would be helpful to the reader.

Why does Ala give the largest difference in pH? It's intuitive that Pro would be the worst, but not that Ala would be the best. The Soret coefficients reflect the chemical properties of the molecules, right? So it would be helpful to the reader to explain things a bit more chemically rather than just stating Soret coefficients.

Thermophoresis of short RNA or DNA strands is well known, given the abundance of experimental data. The thermophoretic movement of small molecules used in our experiments, however, are very tedious to measure. Therefore, we do not yet have a full set of physical mechanisms which drive the thermophoresis of amino acids, nucleotides or phosphate.

While the pK of buffer solutions and solutions of amino acids ensures a sufficient buffer capacity for a given pH, their difference in Soret coefficient of proton donors and acceptors is responsible for the formation pH of gradients. Here, steep proton gradients result from large differences of individual Soret coefficients S_T (see Fig. 4b). In water, these Soret coefficients, supposedly depend on following contributions: non-polar solvents (nonionic interactions), thermoosmosis, diffusiophoresis and Seebeck effect (explicit forces) and free energy of ionic shielding (local equilibrium)[Eslahian KA, Majee A, Maskos M, Würger A (2014) Specific salt effects on thermophoresis of charged colloids. *Soft Matter* 10:1931.; Reichl M, Herzog M, Götz A, Braun D (2014) Why Charged Molecules Move Across a Temperature Gradient: The Role of Electric Fields. *Phys. Rev. Lett.* 112.]. Here, the theoretical mechanism of thermophoresis is not yet fully understood. The complexity of the Soret coefficient, however, makes it almost impossible to relate it on a specific set of chemical properties.

We now write in the manuscript: "Our models indicate that the thermophoretic properties of each buffer species are the major source for differences in the proton gradients. However, the thermophoretic properties of small molecules are rather tedious to measure and not all available, making it difficult to give a detailed prediction for individual buffers [Wang, Zilin, Hartmut Kriegs, and Simone Wiegand. "Thermal diffusion of nucleotides." *The Journal of Physical Chemistry B* 116.25 (2012): 7463-7469].

"The inversion of the pH gradient is explained by a negative difference in the Soret coefficient $\Delta S_T = S_T \text{ acceptor} - S_T \text{ donor}$, which is the result of additional non-ionic contributions to thermophoresis."

What are these "additional non-ionic contributions?" What are the implications of longer peptides inverting their distribution?

The nonionic contribution S_T^{NI} of the Soret coefficient S_T (see equation S4) scales linearly with the polymer length, as a result of local, molecule-solvent interaction around the tube-like polymer. For DNA, this was shown as by Maren Reichl et. al. (see Figure, "Why charged molecules move across a temperature gradient: The Role of electric fields." *PRL*, 2015). Here, the contribution of the Soret coefficient which depends on the protonated state S_T^{CM} (see equation S4) becomes comparatively smaller for the longer peptides. As a result, the difference in Soret coefficient ΔS_T between the proton acceptor and donor decreases or is even inverted for longer peptides.

We now clarify this role of non-ionic thermophoresis in the manuscript by writing: "The non-ionic contribution S_T^{NI} of the Soret coefficient S_T (see equation S4) scales linearly with the polymer length, as a result of local, molecule-solvent interaction around the tube-like polymer. (*) In the case of longer peptides, the contribution of the Soret coefficient which depends on the protonated state S_T^{CM} becomes comparatively smaller. As a result, the difference in Soret coefficient ΔS_T between the proton acceptor and donor decreases or is even inverted for longer peptides."

In addition, we also elaborate on the implications of an inversed pH gradient. Here we write in the manuscript: “The inversion of the pH gradients, facilitated by longer peptides, allows for a microscopic additional chemical feedback from polymerization back to the pH gradient reactions to take place in these chambers. Now, long For example, molecules which are that are concentrated at the bottom of the chamber by the accumulation mechanism, remain at low pH values for prolonged periods if the pH gradient is inverted.”

(*) Reichl, M., Herzog, M., Götz, A. & Braun, D. (2014) Why Charged Molecules Move Across a Temperature Gradient: The Role of Electric Fields. *Phys. Rev. Lett.* 112, 198101

Stadelmaier D, Köhler W (2009) Thermal Diffusion of Dilute Polymer Solutions: The Role of Chain Flexibility and the Effective Segment Size. *Macromolecules* 42:9147–9152.

Wang Z, Afanasekau D, Dong M, Huang D, Wiegand S (2014) Molar mass and temperature dependence of the thermodiffusion of polyethylene oxide in water/ethanol mixtures. *J. Chem. Phys.* 141:064904.

Why do larger molecules cycle throughout the chamber and smaller molecules gets stuck at the bottom?

The thermally driven accumulation mechanism is a nonlinear function of the chamber width. For thinner chambers of $< 150\mu\text{m}$ the trend is clear towards an exponentially better accumulation of longer DNA/RNA oligonucleotides (see Figure 5b from Baaske, Philipp, et al. "Extreme accumulation of nucleotides in simulated hydrothermal pore systems." *PNAS*, 2007). The chamber width of $170\mu\text{m}$ in our experiments is optimized for the accumulation of fast diffusing, small molecules. Here, wide pores exhibit higher convection velocities which are necessary to counterbalance the fast diffusion. As a result, the accumulation mechanism concentrates small molecules at the bottom of the pore. In contrast, the combination of both - increased convection velocities and low diffusion - reduces the accumulation efficiency and shuttles longer oligonucleotides through the chamber. Further details on width dependent accumulation characteristics are also depicted in Supplementary Fig. S13, Given chamber widths of 130, 170, and $230\mu\text{m}$ exhibit the highest accumulation efficiency for oligonucleotide lengths of 14, 4, and 1 nt, respectively.

We make this point now more clear in the revised manuscript by adding stochastic trajectories for short and longer RNA strands in Fig. 5b. In addition, we now write: “Chamber widths above $150\mu\text{m}$ for Earth’s gravity field are optimal for the accumulation of fast diffusing, small molecules to achieve a maximum pH gradient. These pores exhibit high convection velocities which are necessary to counterbalance the fast diffusion of the molecules. As a result, the accumulation mechanism concentrates small molecules at the bottom of the pore. Longer oligonucleotides cannot compensate for the convection, reducing their accumulation efficiency while simultaneously being more prone to shuttle through the thermal habitat.”

There is no data to confirm the simulations of oscillatory, convectively driven RNA cycling between different pH regions of the chamber. Demonstrating that the described system actually works in the lab would greatly improve the impact of the paper. Presumably, it shouldn't be so difficult to monitor this with fluorescently labelled RNA molecules.

The oscillatory behavior of RNA as well as other biomolecules is mainly driven by the convective motion inside the chamber. Here, the convection overlays with the concentration profile maintained by the accumulation mechanism. While the convection can be visualized for single particles (e.g. using polystyrene beads), the accumulation of RNA/DNA oligonucleotides has only been shown in bulk experiments.(*). It is unfortunately close to impossible to track single labeled RNA molecules in our experimental setup - the usage of long working distance optics, the thickness of the chamber and the fast diffusion does not allow to see these molecules.

We make this point more clear by adding experiments that monitor the oscillatory movement in the thermal chamber by using polystyrene beads (see Fig. 5a and Supplementary Movie 2). This data shows that convection movement in the chamber shuttles molecules between region of varying pH, prompting them to undergo pH oscillation. The measurements confirm fully our assumptions on the convection flow and is now shown in Figure 5a,b alongside with simulations of shorter and longer RNA for comparison.

We now write in the manuscript: “Laminar convection induces an upwards and downwards movement of molecules inside the chamber. The convective flow is visualized with 1 μm polystyrene beads over the time of 2 min (see Supplementary Movie 2). **b**, Single molecules are shuttled in the existing pH gradient, undergoing subsequent pH oscillations. The stochastic trajectories of a 5, 10 and 200 mer RNA or DNA strand are depicted in black.[...]”.

What happens if mixtures of amino acids are present in the chamber? Would the pH gradient be lost? Since mixtures of molecules were likely present on prebiotic Earth, it is important to understand how robust the system is to mixtures.

First of all, we have not yet made it clear enough in the manuscript that the concentration of buffer molecules does not affect the pH gradient. The buffering agents must be of course concentrated sufficiently high to act as a buffer against bases/acids. This is seen for example from the experiments for phosphate where the concentration does not much affect the created difference in pH (see Supplementary Fig. S6).

We write now in the main text: “Given sufficiently high buffer concentrations, the pH is only affected by the ratio of proton acceptors to donors, and not by their total concentrations.”

We also have found that mixtures of two buffer solutions lead to the formation of an averaged pH gradient. This was exemplary shown by superposing the phosphate buffer with Arginine for differing buffer concentrations (see Supplementary Fig. S8). However, the accumulation mechanism is affected by the thermophoretic movement, which in turn depends on the exact ionic composition of the solution (see equation S4-6). Therefore, mixtures of various buffers including solutions of amino acids lead to the formation of an averaged pH gradient that is shifted by the ionic composition of the solution.

We now write in the main text: “Mixtures of various buffer or amino acids are expected to form an averaged pH gradient, derived from eq.2 and seen on the example for phosphate/arginine mixtures (see Supplementary Fig. S8). This pH gradient, however, is shifted by the exact ionic composition of the solution (see equation S4-6).”

Although errors bars are reported (which is good), there is no indication of the number of replicates.

Typically, we performed the measurements in duplicates, sometimes in triplicates. We now write in the manuscript: “Depicted error bars are given from duplicate measurements ($n = 2$), and in particular cases from triplicates.”

An argument is made that the pH gradient could be exploited by a protocell, but no clear explanation of how is made.

We describe the formation of pH gradients across protocellular membranes in more detail in the discussion paragraph. Here, we wrote: “An exciting prospect of this work is the periodical formation of proton gradients across vesicles. The leakiness of the first membranes, made of fatty acids or phosphorylated isoprenoids, ensures a short-term pH equilibration with its surrounding. Based on the convective flow, vesicles are transported into different pH regions while preserving the equilibrated pH value at the inside, forming pH gradients across the membrane. These pH gradients could be harnessed by primordial forms of the ATP synthase to drive an energy storage system.”

Reviewer #2 (Remarks to the Author):

PROTON GRADIENTS EMERGE FROM HEAT FLOW AT THE MICROSCALE

Keil et al.

This is a rigorous and convincing demonstration that temperature gradients concentrate solutes differentially and produce pH gradients in closed systems. The results are clear and the analysis sound to the best of my understanding.

Given the importance of this process for the origin of metabolism the paper is of wide importance.

I do not require it, but given the broad range of phenomena mentioned in the introduction, the authors might wish to mention plant cell vacuoles as well as lysosomes.

We thank the referee for his suggestions. We rewrote the initial paragraph and included lysosomes and cell vacuoles as examples for specialized cellular compartments. We now write: “Regions of different pH are essential for various cellular processes such as the synthesis of DNA, RNA, and proteins as well as their degradation and recycling. These processes are accommodated in specialized compartments separated by biological membranes and important in lysosomes, mitochondria, and cell vacuoles.[...]”

The authors might also form a view on the possible relevance of the exergonic reactions of mitochondrial respiration forming temperature gradients within cells:-

Chrétien et al. 2017 Mitochondria are 1 physiologically maintained at close to 50 °C. bioRxiv doi: <http://dx.doi.org/10.1101/133223> PDF attached

Based on physical principles – and after a thorough discussion with Seiichi Uchiyama about one of the first studies of this type (Okabe, Kohki, et al. "Intracellular temperature mapping with a fluorescent polymeric thermometer and fluorescence lifetime imaging microscopy." *Nature communications* 3 (2012): 705.) - in my opinion these thermal gradients cannot exist since the power density of biology is by several orders of magnitude too weak to create microscale heating. For our laser driven experiments, we require mW of laser power for a volume of 50x50x50 μm to increase the temperature by 10 K – Those power densities cannot be generated by the mitochondrial respiration.

Small points.

I would somehow specify “across biological membranes” in the first sentence (line 14). pH gradients occur on a macro scale, too, in animals and plants, and it might help the general reader to focus from the beginning on cells, membranes and membrane compartments.

We rewrote the initial paragraph of the manuscript, focusing more on the role of pH gradients in cellular compartments. We now write in the manuscript: “Regions of different pH are essential for various cellular processes such as the synthesis of DNA, RNA, and proteins as well as their degradation and recycling. These processes are accommodated in specialized compartments separated by biological membranes and important in lysosomes, mitochondria, and cell vacuoles.[...]”

Line 26 “hold” should be “holds”.

This has been changed according to your suggestion.

Reviewer #3 (Remarks to the Author):

The paper builds on the authors' prior work describing thermophoresis effects on molecules in confined geometries ("pores"), which is a body of work in the greater goal to understand conditions that allowed biochemical reactions in the early formation of life. The authors are leading this area and have firmly established the effect for concentration of biomolecules and particles of widely varying size and properties, and they have shown how these conditions affect some biochemical reactions including synthesis of long nucleic acids. The current manuscript makes a significant advance beyond prior work by demonstrating formation of stable pH gradients – alongside biomolecule transport/concentration, pH gradients could provide pH cycling conditions that enabled synthesis reactions before the origin of cells with active transport mechanisms. While in some ways this work is a natural evolution in the research of the authors, elucidation of pH gradients in a plausible natural system (heated pore in a rock) is a fundamentally important advance in the grand scheme. In addition, the work is high of quality and complete (with a few small suggestion below).

Main questions:

1. Throughout this paper, long DNA is less confined/trapped than short oligonucleotides (3-10 nt). This seems opposite of findings in the authors' previous work (e.g., Figure 2 of "Extreme accumulation of nucleotides in simulated hydrothermal pore systems") where it appeared that longer DNA was confined/trapped more effectively or as effectively. The authors should comment on this apparent difference in findings.

We thank the reviewer very much for the thorough and positive assessment of our manuscript as well as the supportive comments.

The thermally driven accumulation mechanism relies on the convective flow, which scales quadratically with the width of the flow chamber. Low convective flows thereby achieve the strongest confinement of long molecules, given chamber widths of 70-100 μm . This was recently shown experimentally for 80-200 mers oligonucleotides, demonstrating an efficient accumulation and, in open pore systems, also a length selective behaviour [Kreysing, Moritz, et al. "Heat flux across an open pore enables the continuous replication and selection of oligonucleotides towards increasing length." *Nature chemistry* 7.3 (2015): 203-208].

The accumulation of single nucleotides and small molecules such as formamide has been shown in theory using wider chambers of 150-180 μm [Baaske, Philipp, et al. "Extreme accumulation of nucleotides in simulated hydrothermal pore systems." *Proceedings of the National Academy of Sciences* 104.22 (2007): 9346-9351; Niether, Doreen, et al. "Accumulation of formamide in hydrothermal pores to form prebiotic nucleobases." *Proceedings of the National Academy of Sciences* 113.16 (2016): 4272-4277]. Here, high convection velocities can compensate for the fast diffusion of small molecules. Experimental studies for this are lacking as small molecules cannot be fluorescently labeled without changing their chemical properties. Other detection methods for small molecules involve an interferometric approach by observing the change in refractive index. This method however is restricted to shallow temperature gradients, as temperature fluctuations also modulate the refractive index [Wang, Zilin, Hartmut Krieger, and Simone Wiegand. "Thermal diffusion of nucleotides." *The Journal of Physical Chemistry B* 116.25 (2012): 7463-7469]. Interestingly, the formation of a pH difference might help to provide more insight into the accumulation behavior of small molecules.

We now make this point more clear by writing: "Chamber widths above 150 μm for Earth's gravity field are optimal for the accumulation of fast diffusing, small molecules to achieve a maximum pH gradient. These pores exhibit high convection velocities which are necessary to counterbalance the fast diffusion of the molecules. As a result, the accumulation mechanism concentrates small molecules at the bottom of the pore.

Longer oligonucleotides cannot compensate for the convection, reducing their accumulation efficiency while simultaneously being more prone to shuttle through the thermal habitat.”

More information is provided in the aforementioned question of referee 1: “Why do larger molecules cycle throughout the chamber and smaller molecules gets stuck at the bottom?”

2. Flow conditions and modeling. The Supplementary Material shows that the model was done as three consecutive simulations, but it is not clear how the first two simulations (thermal and fluid flow) can be de-coupled. Specifically, the thermal distribution should be affected by convective heat transport (flow), and the temperature-induced density differences (thermal) should be required to model the resulting fluid flow. In addition, it would be helpful to have brief comment on the scaling of buoyance-driven flow compared to the dimensions/properties used here (e.g., via Rayleigh number).

The convection speed on the order of micrometers per second is not fast enough to affect the thermal distribution at both reversal points. We see the effects only for convection velocities on the order of millimeters per second. Also, the experiments were performed in chambers with a cross-sectional aspect ratio of 47, minimizing the modulation of the resulting fluid flow from local temperature perturbations at the reversal points. In these experiments, a consecutive simulation of thermal and fluid flow yields the same result compared to the coupled simulation, however decreases the computational time. The Rayleigh number for this system is given by $Ra_L = 2$. The scaling of the convection speed is as expected diffusion-based, and exhibit a quadratic increase in velocity upon an increase in chamber thickness, i.e. the convection speed is 4x if the chamber thickness is doubled.

We now write in the Supplementary Materials: “The μm -sized width and a moderate temperature gradient lead to a Rayleigh number of $Ra_L = 2$. As a result, the heat flow as well as the temperature profile almost entirely corresponds to heat transfer by conduction. The heat transfer caused by convection are negligible. Finite-element simulation of the heat transfer and fluid flow were preformed consecutively to decrease the computational time, however, yielded the same results compared to a coupled simulations of heat transfer/fluid.”

3. The paper presents a simplified model and notes that a more complete model was developed – for all simulated results, it should be clarified which model was used. Also, the simplified model can be useful for understanding the key parameters in a simpler context – if the simulation results presented are for the full model, it would be useful to comment on how the simplified model relates (Does it give similar predictions for points that can be related to the simulations? Is the simplified model valid across the range of conditions reported? What are the limitations in range of applicability of the simple model?)

The simulation results presented in the manuscript are based on the time dependent finite-element simulation. The simplified model derives the pH difference at steady-state, by accessing the maximum concentration at the bottom. As a result, the simplified model can neither calculate the time-dependent formation of a pH difference, nor evaluate a pH profile along the height of the chamber. In addition, the simplification of the experimental prefactor $k_{\text{acceptor}} \sim k_{\text{donor}}$ between Eq. 6 and 7 does not hold for large differences in the Diffusion coefficient between the proton donor and acceptor. The simplified model is applicable for shallow proton gradients and is in good agreement with experimental data such as the formation of pH gradients in solutions of phosphate. Here, the model predicts a proton gradient of $\Delta\text{pH} = 0.40, 0.90, \text{ and } 1.2$ compared to experimental values of $\Delta\text{pH} = 0.34, 0.69, \text{ and } 0.93$ for a given temperature difference of $\Delta T = 8, 12 \text{ and } 14 \text{ K}$, respectively.

The simplified model, however, neglects the build-up of an electrical field that leads to an electrophoretic movement of ions, inhibiting the formation of a pH gradient. As a result, the model is not applicable for steep proton gradients, for example formed by large differences in temperature or by large differences in Soret-coefficients of proton donor/acceptor. Here, the predicted pH difference is much higher than experimentally measured. For example, the simplified model derives a proton gradient of $\Delta\text{pH} = 4.37$ for a temperature

difference of $\Delta T = 30$ K, 2-times higher than experimentally measured. The same applies to calculations on the formation of proton gradients in solutions of amino-acids. Here, the simplified model is not applicable as large differences in the Soret-coefficient between proton donor and acceptor result in steep pH gradients.

We now write in the manuscript: “The simplified model is applicable for shallow proton gradients and is in good agreement with experimental data such as the formation of pH gradients in solutions of phosphate. Here, the model predicts a proton gradient of $\Delta\text{pH} = 0.40, 0.90,$ and 1.2 which describes comparably well the experimental values of $\Delta\text{pH} = 0.34, 0.69,$ and 0.93 for a given temperature difference of $\Delta T = 8, 12$ and 14 K, respectively (Fig. 3c). The pH model (Eq. 8), however, neglects the build-up of an electrical field that leads to an electrophoretic movement of ions, inhibiting the formation of a pH gradient. As a result, the model is not applicable for steep proton gradients, for example formed by large differences in temperature or by large differences in Soret-coefficients of proton donor/acceptors.”

In addition, we now clarify which model we used in the manuscript by writing: "Simulation results are derived using the time dependent finite-element model with a commercial solver."

Minor questions/clarifications:

4. Line 47: “thin sheet of water”. This phrase might imply a film with an open face, rather than an enclosed volume – it would be better to reword this for accuracy and also include the essential elements of a temperature gradient.

We now write: “Thermal energy, the sole energy source necessary for the mechanism, triggers an accumulation mechanism for individual ionic species. The heat flow forms a temperature gradient, spanning across sub-millimeter sized water-filled compartments.”

We also changed this in the Abstract by writing: “[...] across a water-filled chamber form and sustains [...]

5. Line 47: “this elementary and abundant setting”. It seems that the authors intended this to refer specifically to a pore with a thermal gradient, but it is not clear. It may be helpful to describe why this condition is elementary and abundant – it is explained further below, but it is not explained here.

We now write in the manuscript:”This is likely a common setting on early earth, found for example in geothermally heated porous rocks such as hydrothermal vents or cooling volcanic sites.(*). This elementary setting [...].”

(*) Russell, Michael J., et al. "On Hydrothermal Convection Systems and the Emergence of Life." *Econ. Geol.* 100.3 (2005): 419-438.

Kelley, Deborah S., et al. "A serpentinite-hosted ecosystem: the Lost City hydrothermal field." *Science* 307.5714 (2005): 1428-1434.

6. Line 50: “these characteristics now include....”. It was not clear to me (until looking at the references) if this was referring to work that had already shown pH gradients created by thermal gradients in pores (like the work here) or some other work. It looks like these articles describe the role of pH gradients in early biochemical reactions, but they are created by an “un-natural” laboratory method, so the meaning of this sentence should be clarified.

Here, we show that the formation of pH gradients is facilitated in a thermally driven system. The same mechanism has previously been shown to feature characteristics such as shifting polymerization reactions towards longer DNA/RNA strands or the selective replication of longer polymers. In combination, these characteristics potentially enable the onset of molecular evolution in a thermal habitat. The references were

intended to connect the role of energy fluxes with biochemical reactions but may be misleading. They were therefore deleted.

For clarification, we now write in the manuscript: “This study now extends these characteristics to include the formation of a stable pH gradient, providing another mechanism for a promising long-term microhabitat for the onset of molecular evolution.”

7. Line 58: “larger or less concentrated molecules or vesicles”. Meaning and origin of the statement is not clear – “larger” I can understand (from the results here), but I don’t understand “less concentrated” in this context (less concentrated small molecules?).

The accumulation of small molecules forms proton gradients given their initial concentration is sufficiently high to act as a buffer against bases/acids. At low concentrations however, small molecules do not alter the pH gradient.

We make this point now more clear in the revised manuscript: “As indicated by the modelling, the formation of proton gradients is not affected by smaller concentrations of larger biomolecules or vesicles.”

8. Line 87: “diffusion” to “diffusion coefficient”

We have changed this as suggested.

9. Line 93: it may be helpful to briefly note the most relevant parameters included in the experimental prefactor.

We now write: “[...], and the experimental prefactor κ_i ranging between 0 and 0.42 (see equation S2,3). The experimental prefactor κ_i is calculated for each ionic species and depends on experimental parameters such as the temperature difference ΔT , the width of the chamber w and the diffusion coefficient of the species D .”

10. Figure 2. Scale bar should be included, and it is not clear what region of the chamber are shown in each colored panel.

This has been changed according to your suggestion. In addition, the height of each region is now also specified.

Missing temperature units for 30 near end of caption.

We added the missing temperature unit.

11. Line 180: “4c). Here, hydronium ions are released at the bottom of the flow chamber while simultaneously reducing them at the top.”

We now write in the manuscript: “Here, the accumulation of proton donors is much stronger compared to proton acceptors, resulting in a deprotonation reaction of proton donors that releases hydronium ions at the bottom of the chamber.”

12. Line 182: “which is the result of additional non-ionic contributions to thermophoresis.” It is not clear what this means.

We now clarify the role of non-ionic thermophoresis in the manuscript by writing: “The non-ionic contribution S_T^{NI} of the Soret coefficient S_T (see equation S4) scales linearly with the polymer length, as a result of local, molecule-solvent interaction around the tube-like polymer.(*). In the case of longer peptides, the contribution of the Soret coefficient which depends on the protonated state S_T^{CM} becomes comparatively

smaller. As a result, the difference in Soret coefficient ΔS_T between the proton acceptor and donor decreases or is even inverted for longer peptides.”

(*) Reichl, M., Herzog, M., Götz, A. & Braun, D. (2014) Why Charged Molecules Move Across a Temperature Gradient: The Role of Electric Fields. *Phys. Rev. Lett.* 112, 198101

Stadelmaier D, Köhler W (2009) Thermal Diffusion of Dilute Polymer Solutions: The Role of Chain Flexibility and the Effective Segment Size. *Macromolecules* 42:9147–9152.

Wang Z, Afanasenkau D, Dong M, Huang D, Wiegand S (2014) Molar mass and temperature dependence of the thermodiffusion of polyethylene oxide in water/ethanol mixtures. *J. Chem. Phys.* 141:064904.

13. Supplementary Information: the models results are included in some figures but not in all – there is plenty of verification of results here so it is not a critical point, but if there is a reason models were not included in some cases, it would be good to comment. If not, it would further strengthen the work to include them.

The finited-element simulation is in good agreement with our experimental data, shown for various buffer solutions and experimental settings. Here, based on the reaction kinetics and shape of the curves for experimental data and simulation, we do not expect a contradiction from our theoretical model. In addition, the fitting routine of the finite element model for experimental data turned out to be time-consuming. Thus we focused the simulations on experimental results shown in the main text.

14. Figure caption 4c: “c, Peptides have the tendency to form an inversed proton gradient, showing a decrease of pH at the bottom.” Is this generally true or only for the specific peptides observed?

The inversion of a proton gradient has yet only been detected for these specific peptides. Based on physical nature of thermophoresis, however, we also expect it to apply for other peptides as well. Details on the thermophoretic movement of these peptides are elaborated in aforementioned point 12 as well as in the question of referee 1 (What are these "additional non-ionic contributions?" What are the implications of longer peptides inverting their distribution?).

We now write in the manuscript: “Measured peptides have [...]”

15. Results section for Figure 5 (around lines 198-220): It seems that the text is out of line with figure order, and some references to figures seem incorrect.

- “Stochastic simulations show that 10-mer strands of RNA complete pH oscillations of at least $\Delta\text{pH} = 0.5, 1.0, \text{ and } 1.5$ in 11, 38 and 73 min, respectively (Fig. 5b).” – intended to refer to Figure 5d or 5c?
- “As a result, longer DNA/RNA strands such as 200mers undergo faster pH oscillations of at least $\Delta\text{pH} = 0.5, 1.0, \text{ and } 1.5$ in 7, 14, and 26 min, respectively.” – intended to refer to Figure 5c?
- “The accumulation mechanism not only defines the magnitude of pH oscillations for a species but also affects the probability for a species to remain in a specific pH region (Fig. 5a).” – Figure 5b??
- “The flow chamber efficiently accumulates DNA/RNA strands in the region of 3-10 nt, while smaller and larger molecules accumulate significantly weaker (Fig. 5c).” – Figure 5b??
- End of paragraph, and Figure 5 caption – intended to refer to Fig S13, not S12?

We have corrected this as suggested. We also added the oscillatory movement in the thermal chamber by using polystyrene beads in Figure 5 alongside with simulations of shorter and longer RNA for comparison.

REVIEWERS' COMMENTS:

Reviewer #1 (Remarks to the Author):

I am satisfied with the response to my questions and with the modifications to the text. The work is good and relevant to the origin of life field.

I guess what probably wasn't clear from my comments last time was that it is hard for me to see how this paper is significantly different from previous papers in the field. They have already shown that thermal gradients result in gradients of molecules. The only difference now is that these molecules have pKa's that further result in a pH gradient. Maybe the authors would like to clarify this in the text.

Reviewer #3 (Remarks to the Author):

Most of my questions have been answered, but I am still unclear on one of the key questions I had originally (Reviewer #3, Comment #1 - long DNA is less confined/trapped than short, but seems opposite of findings in previous paper). After the response I still do not understand, and the text added to the paper seems confusing still. The response from the authors mostly discusses small molecules other than DNA, so I think there was some misunderstanding of my question - I was not referring to relative concentration of DNA versus non-DNA small molecules, rather the relative concentration of long DNA versus short DNA, where there seems to be a discrepancy with the authors' previous work.

Figure 5b in "Extreme accumulation..." shows that 1000 bp DNA has a higher concentration factor than 100 bp or 1 bp, and this appears to be true by a large magnitude over the full range of pore widths considered (which covers the range of 150 um and 170 um discussed in the response) - yes, the 100 bp and 1 bp are similar for the larger pores, but it does not look like the small DNA is ever concentrated MORE than the long DNA. From this, I would not expect the longer DNA strands in Figure 5B (new paper) to show less accumulation than the short ones.

Similarly, the statement in the paper seems to address a different point - I understand the statement that fast convection is able to overcome the diffusion of small molecules. But I do not understand "Longer oligonucleotides cannot compensate for the convection" - this is too vague and "reducing their accumulation efficiency while simultaneously being more prone to shuttle through the thermal habitat" seems to contradict the results in the previous paper. Similarly, several original statements do the same - "Short DNA/RNA strands accumulate more strongly at the bottom.... Longer strands, however, are more likely to be shuttled..." in the text and figure caption.

I understand that these points only affect a small part of the paper and do not undermine the importance of the work, but since these statements are made directly as general facts, and seem contradictory, they should be addressed.

We thank the referees for the useful comments and hope to clarify the manuscript with the following changes:

Reviewer #1 (Remarks to the Author):

I am satisfied with the response to my questions and with the modifications to the text. The work is good and relevant to the origin of life field.

I guess what probably wasn't clear from my comments last time was that it is hard for me to see how this paper is significantly different from previous papers in the field. They have already shown that thermal gradients result in gradients of molecules. The only difference now is that these molecules have pKa's that further result in a pH gradient. Maybe the authors would like to clarify this in the text.

Albeit the overall mechanism of thermophoretic accumulation has been shown before, we could demonstrate in this work that the timescale of thermal accumulation for small buffer molecules is fast enough to shift the equilibrium state between the buffer and the very diffusive hydronium ions. This is a significant advance compared to previous work that mostly focused on biochemical reactions between larger molecules (e.g. PCR/DNA) or unreactive short molecules (e.g. formamide).

We clarify this issue in the text by now writing:

"This elementary setting has previously been shown to concentrate dilute nucleotides, accumulate lipids to facilitate the formation of vesicles, shift polymerization reactions towards longer DNA/RNA strands, and selectively replicate longer polymers. This study now extends these characteristics to include the formation of a stable pH gradient by thermally separating dissolved buffer molecules of different charge states. This locally shifts the buffer equilibrium, yielding pH differences of up to two units and provides another aspect for a promising long-term microhabitat for the onset of molecular evolution."

Reviewer #3 (Remarks to the Author):

Most of my questions have been answered, but I am still unclear on one of the key questions I had originally (Reviewer #3, Comment #1 - long DNA is less confined/trapped than short, but seems opposite of findings in previous paper). After the response I still do not understand, and the text added to the paper seems confusing still. The response from the authors mostly discusses small molecules other than DNA, so I think there was some misunderstanding of my question - I was not referring to relative concentration of DNA versus non-DNA small molecules, rather the relative concentration of long DNA versus short DNA, where there seems to be a discrepancy with the authors' previous work.

Figure 5b in "Extreme accumulation..." shows that 1000 bp DNA has a higher concentration factor than 100 bp or 1 bp, and this appears to be true by a large magnitude over the full range of pore widths considered (which covers the range of 150 um and 170 um discussed in the response) - yes, the 100 bp and 1 bp are similar for the larger pores, but it does not look like the small DNA is ever concentrated MORE than the long DNA. From this, I would not expect the longer DNA strands in Figure 5B (new paper) to show less accumulation than the short ones.

Similarly, the statement in the paper seems to address a different point - I understand the statement that fast convection is able to overcome the diffusion of small molecules. But I do not understand "Longer oligonucleotides cannot compensate for the convection" - this is too vague and "reducing their accumulation efficiency while simultaneously being more prone to shuttle through the thermal habitat" seems to contradict the results in the

previous paper. Similarly, several original statements do the same - "Short DNA/RNA strands accumulate more strongly at the bottom.... Longer strands, however, are more likely to be shuttled...." in the text and figure caption.

I understand that these points only affect a small part of the paper and do not undermine the importance of the work, but since these statements are made directly as general facts, and seem contradictory, they should be addressed.

The reason for the seeming discrepancy between smaller (e.g. 5mer) and larger molecules (e.g. 500mers) are the different time-scales for their thermophoretic accumulation: For the pore width considered in this work, smaller molecules accumulate faster, but less efficient, while larger molecules are more efficiently concentrated on much longer time-scales.

The large molecules, however, remain on their laminar flow lines for a longer time due to their low diffusivity and are only slowly pushed by thermophoresis towards the cooler region of the reaction vessel.

This corresponds to the situation shown in Fig. 5b: The 5mer is concentrated faster than the 200mer. However, the 200mer would be accumulated much more efficient than the 5mer in the steady state of accumulation as shown in the previous publication mentioned by referee #3 ("Extreme accumulation..."). Figure 5b, however, only considers the experimental time scales (3h) and not the steady state of accumulation for long DNA strands which would be reached after days within the used geometry.

In order to clarify this issue, we modified the text accordingly:

"As a result, the accumulation mechanism concentrates small molecules at the bottom of the pore quickly. Longer oligonucleotides cannot compensate for the convection on the considered experimental and simulation timescales. Until they reach their steady state of thermophoretic accumulation after days or even weeks, they shuttle through the thermal habitat and the zones of different pH."

"The flow chamber efficiently accumulates DNA/RNA strands in the region of 3-10 nt, while smaller and larger molecules accumulate significantly weaker within the considered timescales."

In the mentioned figure caption, we now write:

"Short DNA/RNA strands accumulate quickly at the bottom, remaining mostly in a distinct pH region. Longer strands, however, are shuttled between varying pH regions for days until reaching the steady state of accumulation."